# Revolutionizing Brain Tumor Care: Emerging Technologies and Strategies

**DOI:** 10.3390/biomedicines12061376

**Published:** 2024-06-20

**Authors:** Trang T. T. Nguyen, Lloyd A. Greene, Hayk Mnatsakanyan, Christian E. Badr

**Affiliations:** 1Ronald O. Perelman Department of Dermatology, Perlmutter Cancer Center, NYU Grossman School of Medicine, NYU Langone Health, New York, NY 10016, USA; 2Department of Pathology and Cell Biology, Columbia University Medical Center, New York, NY 10032, USA; lag3@columbia.edu; 3Department of Neurology, Massachusetts General Hospital, Neuroscience Program, Harvard Medical School, Boston, MA 02129, USA; hmnatsakanyanmovsesyan@mgh.harvard.edu (H.M.); badr.christian@mgh.harvard.edu (C.E.B.)

**Keywords:** glioblastoma multiforme, fMRI, temozolomide, TTFields, targeted therapy, external beam radiation therapy, gene and cell therapy, immunotherapy, artificial intelligence (AI)

## Abstract

Glioblastoma multiforme (GBM) is one of the most aggressive forms of brain tumor, characterized by a daunting prognosis with a life expectancy hovering around 12–16 months. Despite a century of relentless research, only a select few drugs have received approval for brain tumor treatment, largely due to the formidable barrier posed by the blood–brain barrier. The current standard of care involves a multifaceted approach combining surgery, irradiation, and chemotherapy. However, recurrence often occurs within months despite these interventions. The formidable challenges of drug delivery to the brain and overcoming therapeutic resistance have become focal points in the treatment of brain tumors and are deemed essential to overcoming tumor recurrence. In recent years, a promising wave of advanced treatments has emerged, offering a glimpse of hope to overcome the limitations of existing therapies. This review aims to highlight cutting-edge technologies in the current and ongoing stages of development, providing patients with valuable insights to guide their choices in brain tumor treatment.

## 1. Introduction

GBM, the most aggressive brain tumor, typically leads to a 12–16 months life expectancy. It has a low annual incidence (~3.19 per 100,000 in developed countries) but is rising in some places due to aging populations and better diagnosis. Diagnosis typically occurs around age 64, with a peak incidence (15 per 100,000) between the ages of 75 and 84 years [1]. The traditional diagnosis of GBM is largely based on its histopathological features. The 2021 revision of the WHO classification of central nervous system tumors incorporates more molecular characteristics as part of the definition of gliomas [2,3]. These include cyclin-dependent kinase inhibitor 2A/B (*CDKN2A/B*) homozygous deletion mutation, telomerase reverse transcriptase (*TERT*) promoter mutation, epidermal growth factor receptor (*EGFR*) gene amplification, and combined gain of entire chromosome 7 and loss of entire chromosome 10 (+7/−10) as qualifying for the diagnosis of GBM, isocitrate dehydrogenase (*IDH*)-wildtype [3,4].

Standard care for newly diagnosed patients with GBM typically involves surgical resection followed by a combination of chemotherapy using temozolomide (TMZ) and radiotherapy [5]. However, recurrence often occurs within several months despite these treatments [6,7,8]. The rapid recurrence of GBM is attributed to its biological characteristics, primarily its invasive nature that makes complete removal through surgery, chemotherapy, or radiotherapy challenging [9]. During tumor resection, the primary goal is to remove as much tumor mass as possible while preserving neurological function. This delicate balance is complicated by the difficulty in distinguishing between normal brain tissue and tumor cells, as removing too much normal tissue can lead to cognitive deficits such as memory loss or impaired thinking and reasoning. Despite advancements in MRI contrast imaging, glioblastoma cells can infiltrate deeper into the brain than imaging can detect.

Radiotherapy is commonly employed post-surgery to target residual tumor cells using high-energy beams [10]. However, this can sometimes lead to the formation of dead brain tissue at the radiation site, and while effective, radiotherapy can also cause significant normal tissue toxicity. Short-term side effects may include fatigue, hair loss, or headaches, while long-term effects can manifest as cognitive changes. Chemotherapy is another treatment modality often combined with surgery and/or radiation [11]. However, the blood–brain barrier (BBB) presents a challenge to the efficacy of chemotherapy in brain tumors [12,13]. Despite advances in adjuvant therapies, achieving complete tumor resection remains a critical factor influencing patient prognosis [14].

This review paper aims to explore current and emerging treatment methods for adult brain tumors and evaluate their advantages and disadvantages to provide patients with informed choices regarding their treatment options.

## 2. Imaging

Magnetic Resonance Imaging (MRI) stands as the gold standard for visualizing GBM, utilizing powerful magnetic fields, radio waves, and computer processing to reveal internal structures. To enhance clarity, a contrast medium like gadolinium is administered intravenously. Various MRI techniques aid in brain tumor diagnosis, including native T1-weighted (T1w), contrast-enhanced (T1-CE), T2-weighted (T2w), and T2-fluid-attenuated inversion recovery (T2-FLAIR) sequences. T1- and T2-weighted imaging highlight signal intensity, delineating tumor borders, while FLAIR improves boundary visualization [15].

### 2.1. Intraoperative MRI (iMRI)

iMRI is a sophisticated imaging technique used during brain surgery to provide real-time visualization of the brain’s anatomy [16]. Unlike conventional MRI scans performed before surgery, iMRI allows surgeons to assess and adjust their surgical plans while the operation is ongoing. This technology has revolutionized neurosurgery by enhancing precision and maximizing tumor removal while minimizing damage to healthy brain tissue [17]. The setup for iMRI involves a specialized MRI machine located within or near the operating room [18]. This proximity enables the seamless integration of imaging with the surgical procedure. During surgery, after a portion of the tumor is removed, the patient is transported to the iMRI for imaging. This process allows surgeons to accurately assess the extent of tumor removal and identify any residual tumor that may require further resection [19]. Furthermore, iMRI-guided surgery reduces the risk of complications and shortens recovery times, leading to better overall patient experiences [17].

### 2.2. Functional MRI (fMRI)

fMRI, pioneered in the 1990s by Seiji Ogawa and Kenneth Kwong, tracks brain activity via blood flow and oxygen levels, known as blood oxygenation level-dependent (BOLD) imaging [20,21]. Active brain areas display increased oxygen consumption and blood flow, appearing brighter on fMRI scans [22]. This technology aids in pinpointing brain regions controlling movement, speech, and vision, which are crucial for preoperative planning and evaluating treatment response [23]. A study contrasting 189 newly diagnosed patients with GBM with their healthy counterparts revealed flatter spectra and reduced gray matter fluctuations in fMRI. Notably, MGMT promoter methylation was correlated with these findings, with varying degrees of flatter spectra predicting overall survival.

In task-based fMRI, patients switch between a passive resting state and performing tasks, often related to motor or language functions. Changes in the BOLD signal are measured to identify areas of cortical activation. Task-based fMRI has been validated to localize anatomical areas, closely approximating functional sites pinpointed by cortical stimulation mapping. In addition to pinpointing eloquent cortex regions, task-based fMRI is valuable for characterizing tumors. It detects decreased BOLD signals in the tumor-involved cortex, with distinctions between high- and low-grade tumors, indicating variations in the cerebral blood volume in the affected area.

Recently, resting-state functional MRI (rs-fMRI) has garnered attention in the field of neurosurgery [24,25]. It does not rely on patients cooperating with task paradigms and can be performed under anesthesia [26,27]. Rs-fMRI identifies spontaneous low-frequency fluctuations in the BOLD signal between spatially distinct regions to identify functional networks, known as resting-state networks (RSNs). The primary RSN is the default mode network (DMN), and research is ongoing regarding other RSNs like somatosensory, visual, auditory, language, attention, and cognitive control networks. Compared to task-based fMRI, rs-fMRI can simultaneously identify multiple networks, offering a more comprehensive view of the brain’s functional architecture while reducing the imaging time [28]. While most studies have focused on functional connectivity and cognition, a few smaller studies have explored rs-fMRI’s ability to depict changes in vascular physiology and tumor grade, as well as predict post-surgical neurological changes. While these studies show promise, further research is necessary before rs-fMRI can be routinely used in clinical settings [28].

### 2.3. Quantitative MRI (qMRI)

qMRI techniques measure physical tissue parameters like T1, T2, T2* relaxation times, or proton density, effectively minimizing hardware-related effects [29,30,31]. Unlike standard clinical MRI, qMRI methods typically require obtaining a series of weighted raw images with varied timing parameters, such as different echo or inversion times [32]. This allows for the exponential fitting of the signal’s time course, producing maps of the corresponding parameter. However, this process can result in relatively long acquisition times depending on the number of images in the series [29,33,34].

In the context of assessing brain tumors, qMRI holds significant utility [31]. Firstly, it assists in identifying tumor infiltration beyond the enhanced region of high-grade gliomas, which is crucial as recurrences often arise from these margins, indicating non-enhancing infiltration zones [35,36]. Secondly, qMRI aids in characterizing brain microstructure neuropathologically. Typically, brain tumors exhibit heightened T1 and T2 relaxation times, correlating with lactate dehydrogenase levels and inversely correlating with vessel density. Changes in T1 relaxation times pre- and post-gadolinium contrast administration weakly correlate with Ki67 levels, a cellular marker for proliferation, and negatively correlate with necrosis extent [37]. Thirdly, it helps differentiate between tumor recurrence and post-treatment changes during follow-up.

Moreover, qMRI contributes to predicting overall survival (OS) and progression-free survival (PFS) [38,39]. For example, patients with post-Bevacizumab T2 relaxation times exceeding 160 ms are likely to experience earlier progression, demonstrating statistical significance in both OS and PFS [40]. Similarly, post-contrast T1 subtraction maps correlate with OS and PFS predictions based on reduced tumor volume following Bevacizumab treatment. This volume reduction is directly associated with improved OS and PFS compared to conventional MRI data [41].

### 2.4. Radiomic

In recent years, artificial intelligence (AI) and machine learning have gained popularity among radiologists and data scientists [42]. In terms of image acquisition and post-processing, AI offers several advantages: (1) it can generate high-quality images quickly, which is especially beneficial for patients who are claustrophobic or uncooperative; (2) it can detect and eliminate motion artifacts like magnetic field inconsistencies and inadequate water or lipid suppression; (3) it can reconstruct full-dose post-contrast images from low-dose or zero-dose acquisitions, which is crucial in situations where contrast agents cannot be used; and (4) it can employ algorithms to create images with improved spatial and contrast resolution compared to the originals [43,44,45].

Radiomics is a fast-growing field within radiology, where conventional radiological images are converted into quantitative data that can be analyzed to decode tumor characteristics. These data can enhance diagnosis, predict prognosis, and assess treatment response [46]. In clinical practice, radiomics has shown promise in evaluating genetic mutations in primary tumors, particularly glial tumors. Radiomics models have been developed to predict genetic features like MGMT methylation and EGFR-A289V mutations in high-grade gliomas and ATRX mutations in lower-grade gliomas (LGGs) [47,48].

Radiomics also holds potential in predicting the response to radiation therapy when combined with temozolomide, as the response often hinges on biological variability, making personalized dosing advantageous [49]. Despite its ongoing expansion, radiomics faces challenges related to data standardization, impacting its reliability, reproducibility, and applicability in different settings [50].

In brief, MRI has evolved as a key tool in modern neuroimaging, offering superior structural insights into brain tumors’ cellular, vascular, metabolic, and functional properties. Integrating all MRI techniques is crucial for delivering optimal clinical care at different disease stages.

## 3. Surgery

Brain tumors are often initially treated with surgery, particularly when the risk of neurological damage is minimal. Modern surgical approaches for brain tumors involve a comprehensive strategy that combines advanced imaging techniques, neuro-navigation systems, and intraoperative monitoring methods. Before surgery, patients undergo detailed imaging like MRI and CT scans to precisely identify and characterize the tumor, which is crucial for effective surgical planning [51]. During the procedure, neurosurgeons employ various techniques such as craniotomy, which involves opening the skull to access the brain, and minimally invasive methods like neuroendoscopy or stereotactic biopsy for challenging tumor types or locations [51,52].

Intraoperative techniques are crucial in assisting surgeons in achieving complete tumor resection, especially because distinguishing tumoral tissue from normal brain tissue can be challenging. Commonly used techniques include 5-ALA fluorescence, intraoperative MRI or ultrasound, and navigation systems [17]. In neurosurgery, 5-ALA fluorescence is employed for the intraoperative visualization of malignant brain tumors. Patients ingest 5-ALA orally, which is absorbed and metabolized by tumor cells, leading to the production of fluorescent porphyrins. When exposed to blue light during surgery, these porphyrins emit red fluorescence, aiding surgeons in accurately distinguishing tumor tissue from healthy tissue.

Moreover, the development of intraoperative tools, such as the intraoperative PET probe, holds great promise due to its superior capability to detect tumor recurrences across various tumor types. These PET probes assist surgeons in precisely identifying tumor margins by detecting radioactive tracers that accumulate in cancerous tissues. This guidance enables surgeons to remove as much of the tumor as possible while preserving healthy tissue. Additionally, the post-surgical use of the PET probe can verify the surgery’s success by evaluating the extent of tumor removal or improvements in tissue function.

### 3.1. Morphological Imaging Modalities

Morphological imaging remains essential for guiding resection surgery, with MRI being the gold standard for achieving MRI-based complete resection. Despite practical and financial challenges, intraoperative ultrasound techniques such as B-mode ultrasound and elastosonography have shown effectiveness in defining tumor margins and infiltration in various glioma grades [53,54]. Additionally, tools like optical coherence tomography are promising for identifying vascular structures, particularly in noninvasive tumors like meningiomas, where preserving essential vessels during resection is crucial [55].

### 3.2. Fluorescence

Advanced techniques enable the direct identification of tumor cells using fluorescence. Endogenous autofluorescence, utilized through time-resolved fluorescence spectroscopy, has shown promise in distinguishing glioma cells from healthy tissues in early clinical data [56,57]. While endogenous autofluorescence and exogenous fluorescence-guided surgery using 5-ALA are valuable in clinical practice, the latter stands out due to its sensitivity, specificity, and low phototoxicity [58,59]. Its systematic use in GBM surgery has significantly improved gross total resection rates and is being considered for other tumor types [57,60].

### 3.3. Awake Brain Surgery

Studies in neuropsychology have shown that cognitive and behavioral changes can occur after brain surgery, even in tumors in the right hemisphere [61,62]. The right hemisphere is crucial for motor control, visual processing, spatial cognition, semantic processing, attention, and social cognition [63]. For patients with longer life expectancies, such as those with low-grade gliomas, using awake surgery with cortical and axonal electrostimulation mapping for right-sided tumor resection is recommended [64,65]. Awake brain surgery, also called awake craniotomy, is performed while the patient is conscious and alert. It is used to treat brain conditions near areas controlling vision, movement, or speech. During this surgery, patients may be asked questions, and their brain activity is monitored as they respond, helping surgeons target the precise area needing treatment and reducing the risk of damaging functional brain areas affecting vision, movement, or speech [63,66].

While awake craniotomy is beneficial for preserving language and cognitive functions during surgery, its use may be limited by patient factors like anxiety and cooperation. Future research should explore strategies to optimize patient selection, improve intraoperative monitoring, and minimize potential adverse effects to maximize the benefits of awake surgery while ensuring patient comfort [67,68].

### 3.4. Robotic Surgery

Robotic surgery has emerged as a cutting-edge approach for treating brain tumors with enhanced precision and minimal invasiveness [69,70]. This advanced technology allows surgeons to perform intricate procedures with greater accuracy, particularly in the hard-to-reach brain areas. Robotic systems offer high-definition imaging and real-time feedback, enabling surgeons to navigate complex brain structures with unprecedented clarity and control [71].

One of the key advantages of robotic surgery in brain tumor treatment is its ability to minimize trauma to surrounding healthy brain tissue [71]. The precision of robotic instruments reduces the risk of damage to critical areas responsible for cognitive and motor functions [71,72]. This precision is especially beneficial for tumors located near eloquent brain regions, where preserving functional integrity is paramount. Furthermore, robotic surgery offers patients several benefits, including shorter recovery times, reduced postoperative pain, and lower complication rates compared to traditional open surgeries. The minimally invasive nature of robotic procedures often results in smaller incisions, leading to improved cosmetic outcomes and a quicker return to daily activities for patients [73]. In conclusion, robotic surgery represents a significant advancement in the field of brain tumor treatment, offering superior precision, minimal invasiveness, and improved patient outcomes. Continued research and technological innovations in robotic-assisted procedures hold great promise for further enhancing the effectiveness and accessibility of brain tumor surgery in the future (Figure 1).

## 4. Irradiation

The goals of radiation therapy are to kill tumor cells, slow tumor growth, and ultimately stop the tumor from growing while limiting the amount of radiation to nearby healthy brain tissue and vital organs. Radiation may be used after all or part of a tumor is removed by surgery [74,75]. GBM is a highly invasive cancer, with tumors presenting irregular boundaries and GBM cells infiltrating through the normal brain. Therefore, the complete removal of all tumor tissue is highly complicated. Radiation attempts to destroy the remaining GBM cells [26]. Treatments are painless and take just a few minutes. A typical schedule for radiation therapy consists of one treatment per day, five days a week, for two to seven weeks. The standard radiation dose for treating GBM is 60 Gy in 30 fractions [75,76].

### 4.1. External Beam Radiation Therapy (EBRT)

Intensity Modulated Radiation Therapy (IMRT): is a type of external beam radiation therapy that employs computer-controlled linear accelerators to precisely deliver radiation doses to a tumor or specific areas within it [77]. This technique modulates the intensity of the radiation beams, enabling customized dose distributions that closely conform to the tumor’s shape. By utilizing multiple radiation beams from different angles, IMRT achieves a higher dose to the gross disease while simultaneously sparing healthy brain tissue and radiation-sensitive structures. This approach has demonstrated efficacy in reducing the toxicity associated with radiation therapy [74]. For instance, a study involving 58 patients with high-grade gliomas undergoing IMRT treatment exemplified these benefits. After a 24-month follow-up, they found that PFS was around 5.6 months for anaplastic astrocytoma and 2.5 months for GBM, while OS was approximately 36 months and 9 months, respectively. These results were comparable to historical data from conventional radiotherapy, leading the authors to suggest that IMRT, despite reducing neurotoxicity, may not significantly enhance local control or survival under the current dosage and volume parameters. However, they acknowledged the potential for dosimetric improvements in IMRT to eventually translate into clinical benefits, especially as patient survival increases due to treatment advancements [78].

Volumetric modulated arc therapy (VMAT) is an advanced radiation technique that modulates the dose rate, field aperture, and gantry speed during the delivery of radiation, which creates a more conformal dose distribution, allowing more sparing of normal tissues than compared with IMRT treatment plans [79]. However, while the dosimetric improvements of VMAT when compared to IMRT have not translated to improved clinical outcomes or decreased toxicity, VMAT does result in a mean 29% reduction in treatment time and should be considered as a treatment technique for patients with GBM [79]. VMAT is favored by many health care centers due to its faster treatment delivery and improved sparing of normal tissues compared to IMRT or 3D conformal radiation [79]. While VMAT maintains target coverage goals like IMRT, it offers increased dose conformality and reduced dose to critical areas like the brain stem and optic chiasm. This can lead to better local control and lower radiation toxicity. However, caution is advised when assuming similar outcomes between IMRT and VMAT, as studies show varied results [80]. Despite VMAT’s ability to meet target coverage goals, planning studies suggest the potential for lower doses to the target area. More outcome data are needed to fully understand these differences [81] (Figure 1 and Table 1).

### 4.2. Proton Therapy

Proton therapy stands out as an advanced form of external beam radiation therapy, diverging from traditional photon-based methods (using X-rays) by employing protons for radiation delivery. Protons can be precisely aimed at tumors, concentrating their radiation dose at specific tissue depths, thus minimizing exposure to healthy surrounding tissue. This approach has been explored for its potential to mitigate cognitive deficits and enhance outcomes in patients with GBM [82]. In a phase II trial involving 90 patients with GBM who underwent surgery, they were then randomly assigned to receive radiation therapy at a dose of 60 Gy in 30 fractions alongside temozolomide, either through proton therapy or photon-based radiation. Over a median follow-up period of 48.7 months, no significant differences were observed in the time to cognitive failure (HR = 0.88, *p* = 0.74) or OS (HR = 0.68, *p* = 0.60) between the two groups. A prospective registry study from multiple proton beam facilities in Japan reported a median OS of 21.2 months for patients with GBM treated with proton therapy, deemed comparable to existing treatment outcomes [83]. Additionally, a meta-analysis encompassing trials of proton-based radiation therapy in patients with GBM showed similar clinical outcomes to those seen with photon-based radiation [84]. These findings indicate that while proton therapy is promising, it may not yield significant improvements in the clinical outcomes for patients with GBM (Table 1).

### 4.3. Focused Ultrasound (FUS)

FUS for brain tumors is a groundbreaking technology that offers a noninvasive alternative to traditional surgical procedures [85]. It works by directing sound wave beams to a specific region of the brain using an implanted device or with the aid of real-time magnetic resonance imaging. The combination of ultrasound waves and microbubbles injected into the blood at low frequencies can briefly open the tight junctions between endothelial cells, which typically limit drug delivery to the brain [85]. The key advantage of focused ultrasound lies in its ability to deliver highly concentrated energy precisely to targeted brain areas, minimizing damage to healthy tissue and reducing typical surgery-related complications. This technology is especially beneficial for patients who are ineligible for traditional surgery or those opting to avoid the risks and recovery periods associated with invasive procedures [86]. Moreover, its noninvasive nature often permits outpatient procedures, enabling quicker recovery with fewer side effects [85]. The transient gap created by FUS can facilitate the efficient delivery of drugs, antibodies, and immune cells targeting brain tumors, potentially enhancing treatment efficacy and patient outcomes [87] (Table 1).

In conclusion, the administration of radiotherapy necessitates a delicate balance between ensuring effective tumor control and minimizing harm to healthy tissues. This balance is especially critical for elderly patients, who are more susceptible to conditions such as diabetes and atherosclerosis. They face heightened risks of developing radiation-induced encephalopathy and cognitive decline due to disruptions in the microvasculature [88,89]. Age emerges as a significant factor in radiotherapy-induced brain demyelination, impacting 72.9% of individuals aged 50 and above, in contrast to 39.2% of younger patients. Moreover, the risks of neurotoxicity escalate with higher radiation dosages [90]. It is imperative to tailor radiotherapy protocols meticulously for elderly patients, considering their unique vulnerabilities, to maximize treatment efficacy while safeguarding their overall well-being.

**Table 1 biomedicines-12-01376-t001:** Advantages and disadvantages of modern radiotherapy techniques versus conventional radiation therapy.

	Advantage	Disadvantage
IMRT	Tumor-targeting, sparing normal cells Reduced side effects	Prolonged treatment affects the patient’s comfort and positioning Costlier due to complex planning and delivery Needs special equipment, expertise [91,92]
VMAT	Faster treatment delivery Improved dose conformity Reduced radiation exposure to patients	Complex planning and quality assurance needed Higher treatment costs Limited in some healthcare settings [80]
Proton Therapy	Accurate dosing, minimal harm to healthy tissues Reduced risk of long-term side effects	Expensive to set up and maintain Limited in some healthcare settings [93]
FUS	Noninvasive, no incisions or radiation Tumor-targeting, tissue-safe Quick recovery time and minimal side effects	Restricted to specific tumor locations Effectiveness varies by tumor size and type Limited in some healthcare settings [85,87,94]

Note: IMRT: Intensity Modulated Radiation Therapy, VMAT: Volumetric modulated arc therapy, FUS: Focused Ultrasound.

## 5. Chemotherapy

Epigenetic changes such as DNA methylation, histone modification, and chromatin remodeling are seen as key features in the development and progression of GBM. They have been identified not just as possible biomarkers for effectively categorizing patients in clinical settings but also as potential targets for drugs due to their ability to be reversed. A well-researched case is the methylation status of the MGMT promoter, which has been shown to forecast the effectiveness of alkylating chemotherapy and is clinically significant in predicting outcomes in patients with GBM [95,96].

Temozolomide (TMZ), also known as Temodar, is the primary FDA-approved drug for newly diagnosed GBM. TMZ effectively penetrates the BBB and acts as a DNA alkylating agent [2,97]. TMZ is typically taken seven days a week for six weeks at a dosage of 75 mg/m^2^ during the initial treatment phase. After this, patients undergo six cycles of adjuvant TMZ therapy at a dosage ranging from 150 to 200 mg/m^2^ [8]. Each cycle spans 28 days and includes five days of TMZ administration. TMZ operates by attaching a methyl group to specific guanine positions: oxygen-6 (O6), nitrogen-7 (N7), and adenine-3 (N3), disrupting DNA chains and leading to cytotoxic bases like O6-methylguanine (O6-MG), N7-methylguanine (N7-MG), and N3-methyladenine (N3-MA) [2,97]. These alterations cause DNA replication to stall at mismatched base pairs, inducing cell cycle arrest at the G2/M phase. The O6-methylguanine-DNA methyltransferase (MGMT) gene codes for a protein that repairs alkyl damage to guanine’s O6 position, a key site of DNA alkylation. MGMT expression or its tolerance to mismatch repair-deficient (MMR-) states, confers resistance to TMZ. Patients with MGMT methylation exhibit improved median OS rates compared to those without it: 21.7 months with chemotherapy and radiation versus 15.3 months with radiation alone for patients with methylated MGMT [75,97]; 12.7 months with chemoradiation versus 11.8 months with radiation alone for patients with unmethylated MGMT. Thus, MGMT promoter methylation is a crucial predictor of TMZ chemotherapy efficacy, especially in personalizing treatments for elderly patients [95,96] (Figure 1).

Research has established a disparity in glioblastoma (GBM) survival based on sex, where females exhibit a notable survival advantage. The incidence of GBM is higher in women’s right temporal lobe, while in men, the left temporal lobe is more affected [98,99]. A large study conducted by Dr. Barnholtz-Sloan’s group at the National Cancer Institute found no sex-based survival differences among individuals with MGMT promoter unmethylated and those with MGMT promoter methylated who received either TMZ and radiation or TMZ alone [100]. However, a significant survival gap was observed among individuals with MGMT promoter methylated, particularly with females showing significantly lower survival rates than males when treated with radiation alone (females: 13.4 months vs. males: 24.5 months, *p* = 0.01) [100].

A study in 2015 demonstrated the capability of interferon alpha to reduce MGMT expression, thereby enhancing the effectiveness of TMZ [101]. Interferon alpha, associated with the innate immune system, has immunomodulatory, antiproliferative, and antiangiogenic effects. In a phase III clinical trial, patients receiving interferon alpha alongside TMZ and radiation demonstrated a greater median OS than those receiving only TMZ and radiation (26.7 vs. 18.8 months, *p* = 0.005) [102]. Subset analysis revealed that among patients lacking MGMT methylation, interferon alpha use significantly improved median OS (24.7 vs. 17.4 months). This finding is crucial, as patients without MGMT methylation typically have a poorer prognosis, highlighting interferon-alpha’s potential to enhance clinical outcomes in this subgroup [102].

## 6. Targeted Therapy

Targeted therapy is a strategic approach utilizing medications specifically designed to target molecules or pathways crucial for cancer cell growth and survival, offering a more precise and lower side effect alternative to traditional chemotherapy. To enhance TMZ therapy’s effectiveness, two primary strategies are explored: (1) identifying or developing chemosensitizers that can potentially bypass TMZ resistance or amplify its effectiveness in drug-resistant tumors and (2) creating innovative formulations that bolster the persistence of the drug within the body and specifically within tumors. Moreover, combining TMZ with synergistic targeted drugs not only improves brain tumor treatment efficacy but also reduces overall chemotherapy toxicity, promising better patient outcomes and advancements in cancer therapeutics.

### 6.1. DNA Damage Repair Targeting Drugs

PARP, an enzyme vital for MGMT activation and repairing TMZ-induced DNA damage, has been the focus of several studies. These studies demonstrate that combining PARP inhibitors like veliparib and olaparib can restore chemosensitivity to TMZ in MSH6-inactivated, MMR-deficient GBM cells [26,103]. Several PARP inhibitors (PARPis), such as olaparib, veliparib, niraparib, and pamiparib, are currently being studied in combination with TMZ in patients with GBM. One potential advantage of using PARPis is that some of these compounds have the necessary physicochemical properties to effectively cross both the blood–brain barrier (BBB) and blood-tumor barrier (BTB) (Figure 2).

### 6.2. Histone Deacetylase (HDAC) Inhibitors

Epigenetic alterations related to histones primarily involve histone acetylation and deacetylation processes. Histone acetyl transferase (HAT) adds acetyl groups to histones, while histone deacetylase (HDAC) removes them. HDACs have been implicated in activating pathways leading to chemotherapy resistance in GBMs [104]. Currently, researchers are exploring pan-HDAC inhibitors as potential GBM therapies, often in combination with TMZ [105]. Studies have demonstrated significant efficacy improvements in GBM cells in laboratory and mouse models with orthotopic tumors using this combination [106]. Furthermore, a phase II clinical trial is underway to evaluate the effectiveness of vorinostat combined with TMZ and radiation for newly diagnosed patients with GBM (CT01236560) [105]. Levetiracetam (LEV), a newer antiepileptic drug, regulates HDAC levels to silence MGMT, thus enhancing TMZ’s effectiveness against glioma stem cells (GCSs) [107]. Retrospective analyses and an open-label phase II study (NCT02815410) suggest that LEV can improve PFS and OS in patients with GBM [108] (Figure 2).

### 6.3. Targeting Transcription Factors

By regulating the expression of genes involved in a variety of cellular behaviors, transcription factors play key roles in promoting cancers, including GBM. Among such factors, a growing body of work featuring expression and knockdown studies has identified ATF5, CEBPB, and CEBPD as required elements for GBM formation, growth, cell survival, and treatment resistance [109]. Although transcription factors are often deemed as challenging to target, it was recognized that as members of the “leucine zipper” family, the activities of ATF5, CEBPB, and CEBPD could be compromised by the use of “dominant-negative” peptides [109]. Transcription factors with leucine zippers, including ATF5, CEBPB, and CEBPD, must form dimers to become active and do so via specific dimeric interactions of their helical leucine zipper domains. Thus, their activities may be suppressed by “dominant-negative” peptides that associate with the native leucine zippers but form dimers that cannot bind DNA to directly regulate gene expression [110]. Indeed, the expression of constructs expressing such dominant-negatives in GBM cells, both in vitro and in vivo, triggered their death [109,111]. Importantly, the constructs did not provoke the death of normal, non-transformed cells. To transform the dominant-negative “decoy” peptides into drugs, they were fused with an N-terminal cell-penetrating (CP) penetratin domain, which rendered them capable of passing through tissue barriers and into cells [112,113]. When delivered intraperitoneally, one such peptide (CP-dn-ATF5) was shown to cross the blood–brain barrier, enter intracranial GBM cells, and trigger their death [113,114].

To date, four cell-penetrating peptides have been described in the literature that target CEBPB/CEBPD/ATF5. These are (1) CP-dn-ATF5, which possesses a penetratin domain and a portion of the ATF5 leucine zipper and targets CEBPB and CEBPD [113,114,115]; (2) ST101, a peptide composed of D-amino acids that contains a modified penetratin domain and a leucine zipper sequence that is similar to that in CP-dn-ATF5 and that is described as a CEBPB antagonist [109,116]; (3) Bpep that is composed of a penetratin domain and the CEBPB leucine zipper and that appears to target ATF5, CEBPB, and CEBPD [109,115,117]; and (4) Dpep that has properties similar to Bpep, but contains the CEBPD leucine zipper sequence [109,115,117]. All four peptides have been shown to promote apoptosis of GBM and other tumor cell types while sparing normal cells [109]. Of these, ST101 is currently undergoing clinical trials for advanced solid tumors, including GBM (NCT04478279). While there have been no publications regarding the trial outcomes to date, there have been several abstracts indicating safety and partial responsiveness to the drug.

### 6.4. Cell Cycle Checkpoint Inhibitors

Inhibitors of cyclin-dependent kinase 4/6 (CDK4/6) have shown promise in halting the cell cycle in the G1 phase and effectively inhibiting tumor proliferation [4,118]. Ribociclib, palbociclib, and abemaciclib are among the CDK4/6 inhibitors explored as potential treatments for recurrent GBMs [103,118,119,120]. Notably, abemaciclib demonstrated efficacy in increasing PFS in patients with recurrent GBM in a phase II clinical trial, although it did not impact the OS rates [121]. Conversely, ribociclib and palbociclib, when used alone, were deemed ineffective against recurrent GBM [122]. These results suggest that while CDK4/6 inhibitors could be valuable additions to GBM treatment options, combining them with other cytotoxic drugs may be critical for improving overall survival rates beyond the current standard of care [123] (Figure 2).

### 6.5. Epidermal Growth Factor Receptor (EGFR) Inhibitors

Approximately 50% of malignant gliomas show a mutation in the EGFR [1,2,124]. EGFR dysregulation can increase tumor angiogenesis, migration, and growth, contributing to poorer clinical outcomes in patients with GBM. EGFR-tyrosine kinase inhibitors continue to be a focus of GBM research. While erlotinib alone lacked efficacy in recurrent GBM, a combination with the mTOR inhibitor sirolimus led to a partial response in 19% of the patients. The new selective and highly potent small-molecule EGFR inhibitor with improved CNS penetration, ERAS-801, has received FDA clearance in 2021. ERAS-801 is currently being evaluated in patients with recurrent GBM (NCT05222802) [125]. In patient-derived GBM models, ERAS-801 showed a survival benefit of 93% in EGFR mutant and/or amplified models, significant brain penetrance, and prolonged survival compared to previously approved EGFR-tyrosine kinase inhibitors like erlotinib (Figure 1).

### 6.6. Vascular Endothelial Growth Factor (VEGF) Inhibitors

VEGF plays a pivotal role in promoting angiogenesis in GBM (Figure 2). Bevacizumab, an inhibitor of VEGF, has undergone extensive investigation within the GBM context [126]. In a study focusing on recurrent disease, bevacizumab demonstrated a median PFS of 16 weeks and elicited a radiographic response in 71% of the patients [127]. Consequently, it received FDA approval for recurrent GBM treatment in 2009, both as a standalone therapy and in combination with irinotecan [128]. Nonetheless, the clinical impact of bevacizumab remains somewhat constrained, particularly in the context of newly diagnosed GBM.

Regorafenib, an inhibitor of VEGF that targets multiple VEGF and PDGF receptors, demonstrated a significant improvement in PFS (7.4 vs. 5.6 months) compared to lomustine (another alkylating chemotherapeutic used to treat brain tumors) in patients with recurrent GBM in an initial study (NTC02926222) [129]. Building on these promising initial results, regorafenib underwent further investigation in the GBM Adaptive Global Initiative Learning Environment (AGILE) trial. This international phase II/III platform trial for patients with GBM evaluates various experimental treatments against a common control (TMZ and lomustine), aiming to swiftly identify effective therapies NCT02926222 (Figure 2).

### 6.7. Drug Metabolism Targets

Additional research highlighted a connection between the standard GBM treatment, temozolomide (TMZ), and its impact on fatty acid metabolism. TMZ was found to enhance the uptake of fatty acids, potentially aiding in fatty acid oxidation (FAO) [130]. This alteration in metabolic behavior due to TMZ treatment can lead to therapeutic resistance. To assess therapeutic responses, researchers evaluated the effects of Etomoxir, an inhibitor of CPT1, which mediates fatty acid transfer across mitochondrial membranes, along with TMZ [131]. They conducted these evaluations using patient-derived GBM and orthotopic xenograft mouse models. This reprogramming of tumor metabolism through treatment represents an important mechanism in developing resistance to therapy, suggesting that targeting such metabolic aberrations could be a promising avenue for novel therapeutic strategies. Another study discovered that when TMZ was combined with YTX-7739, an inhibitor of stearoyl CoA desaturase 1 (SCD), animal survival was extended to 136.5 days, compared to 106.5 days with TMZ alone. SCD is an enzyme that converts saturated fatty acids (SFA) into monounsaturated FA (MUFA) [132]. Using YTX-7739 also made these recurrent cells more responsive to radiation therapy [132]. Combining YTX-7739 with standard-of-care DNA-damaging therapy for GBM appears to offer potential benefits by enhancing its therapeutic effectiveness [132,133].

Therapeutically, epigenetic modulators have shown promising results in various cancers, including GBM, and have been investigated in clinical trials as antitumor agents [134,135]. For example, histone deacetylase (HDAC) inhibitor (Vorinostat) has been tested in a phase II study for patients with recurrent GBM [136]. In another discovery, researchers observed that HDAC inhibitors inhibit glycolysis while simultaneously enhancing cellular respiration, fueled by increased in beta oxidation [137]. To capitalize on this finding, Panobinostat was combined with the fatty acid oxidation inhibitor etomoxir. This combination treatment synergistically reduced cellular viability across various GBM cultures, including neurospheres, PDX-derived lines, and established GBM cells [137]. In a separate study, researchers discovered that the combination of Panobinostat and imipridones (ONC201) exerts its effects, in part, by suppressing tumor cell metabolism [138].

### 6.8. Tumor Treating Fields (TTFields) Therapy

TTFields therapy is a noninvasive approach utilizing low-intensity electric fields to target cancer cells, particularly effective due to the higher susceptibility of rapidly dividing cancer cells to disruption compared to normal cells. The Optune System, comprising insulated transducer arrays placed on the skin in the tumor region, generates low-intensity, medium-frequency alternating electric fields (1–3 V/cm, 100–300 kHz). These fields interfere with cell division (mitosis), inducing cell death or inhibiting tumor growth. Moreover, TTFields induce immunogenic cell death (ICD), activate the tumor immune microenvironment, and reduce metastasis and invasion while enhancing BBB permeability [139].

The Optune System, consisting of arrays of insulated electrodes attached to the scalp, continuously generates electric fields via a portable device worn by the patient for at least 18 h daily. Extensively studied in clinical trials, TTFields therapy, especially GBM treatment, demonstrates improved OS and PFS when combined with standard treatments like chemotherapy and radiation therapy [140,141]. Incorporating TTFields alongside maintenance, TMZ demonstrated an increase in the median PFS from 4 to 6.7 months and median OS from 16 to 20.9 months [142]. FDA approval was granted for recurrent GBM in 2011 and for newly diagnosed GBM in 2015 [143].

TTFields therapy also impedes DNA damage repair, enhancing its effectiveness when combined with other antitumor methods [139]. It induces immunogenic cell death, activates various immune cells, and reduces metastasis and invasion by downregulating cytokines and disrupting primary cilia [144]. Additionally, TTFields therapy increases BBB permeability, facilitating the distribution of drugs and immune cells within the tumor. TTFields therapy can be combined with immune checkpoint inhibitors, tumor vaccines, antimitotic drugs, and PARP inhibitors with generally minimal side effects, such as scalp irritation or skin reactions. Offering a well-tolerated alternative to traditional therapies without typical chemotherapy or radiation side effects like nausea or hair loss, TTFields therapy holds significant promise as an addition to cancer treatment options.

### 6.9. Convection-Enhanced Delivery (CED)

In 1994, a team of clinicians and engineers introduced the concept of Convection-Enhanced Delivery (CED), a method for precisely delivering drugs to the brain that are either restricted by the BBB, such as antibodies, nanoparticles, or conjugates (often paired with imaging agents or biomarkers) [145]. CED involves precisely placing one or more catheters using stereotactic techniques directly into the brain parenchyma or tumor under image-guided neuron navigation [146]. These catheters are then linked to pumps that deliver a continuous, positive-pressure micro-infusion of specific agents into the target tissues, utilizing the principles of ‘bulk flow’.

Despite its long-standing use in brain cancer therapy, the widespread adoption of CED in clinical practice has been limited [146]. Various factors, including catheter design, placement, tumor characteristics, infusion parameters, and brain anatomy, influence the effectiveness of CED. Balancing drug distribution within and around tumors while addressing physical limitations is a constant challenge. Fluid dynamics within the brain, particularly interstitial fluid flow, play a significant role in drug transport and can impact disease progression through cellular interactions.

Several studies have explored CED with conventional chemotherapies, like Paclitaxel and Topotecan (TPT), for treating recurrent high-grade gliomas [147]. For instance, Lidar et al. reported positive imaging responses in patients treated with CED of Paclitaxel, with a median OS of 7.5 months [148]. In a small patient cohort, TPT, known for its efficacy in preclinical trials, showed significant antitumor effects and improved survival when delivered via CED in a Phase Ib trial for recurrent high-grade gliomas [16,147]. This innovative method of delivering topotecan addresses challenges in its delivery and the assessment of treatment response in patients with GBM, potentially extending to other anti-glioma medications or even other CNS disorders.

The future of CED lies in refining catheter designs to optimize flow rates, minimize backflow, improve distribution uniformity, and ensure long-term safety and efficacy through innovative materials and features [149,150]. Ongoing research and trials are essential for validating these advancements in clinical practice.

## 7. Gene and Cell Therapy

Gene therapy is a cutting-edge medical strategy employing genetic material, such as DNA or RNA, to address diseases stemming from defective or abnormal genetic conditions. This innovative approach involves modifying targeted genes to prompt the production of proteins that can substitute for defective ones, making it a promising avenue for cancer treatment. Gene therapy shows potential in reducing chemotherapy-induced resistance mechanisms. Two primary strategies in gene therapy target altering tumor cell behavior directly through gene correction and editing, as well as elevating the immune system to identify and eliminate tumor cells, known as immunotherapy and oncolytic virotherapy [151,152].

### 7.1. Immunotherapy

Immune checkpoint inhibitors (ICI) have been widely utilized for treating various types of cancer, such as melanoma, lung, kidney, bladder, and lymphoma. These inhibitors target immune checkpoint proteins found on T cells, which normally send an “off” signal to these cells when they recognize partner proteins in other cells. Tumor cells exploit this mechanism to evade immune system detection. Checkpoint inhibitors function by obstructing the binding of checkpoint proteins to their partner proteins. Monoclonal antibody drugs targeting the primary checkpoint proteins, CTLA-4 (Ipilimumab) and PD-1 (Nivolumab, Pembrolizumab), along with their partner PD-L1 (Avelumab, Atezolizumab, Durvalumab, Cemiplimab), are widely used globally across different cancer types. However, their efficacy in patients with GBM remains limited [153]. These targets present opportunities for inhibiting immune checkpoints, yet the clinical outcomes for GBM have been less encouraging [153]. For instance, Nivolumab (Anti-PD-1) demonstrated minimal to negligible effects across three phase III clinical trials involving diverse patient groups [154]. It is important to highlight that neoadjuvant therapy using anti-PD-1 has displayed encouraging results in specific recurrent patients with GBM within window-of-opportunity trials [103]. An extensive phase II study across multiple centers assessed the Durvalumab combination with standard radiotherapy in newly diagnosed patients with GBM with unmethylated tumors, revealing good tolerability and potential effectiveness. Notably, one patient achieved an impressive OS of 86 weeks [154,155].

Ipilimumab’s efficacy in melanoma has been established, but its impact on GBM is limited to two phase II trials, showing a median OS of up to 7.7 months when combined with anti-PD-1 therapies and few side effects [5]. GBM’s low response to ICI monotherapy may be due to its “immunologically cold” nature, characterized by low lymphocyte infiltration and exhausted lymphocytes within the tumor [156]. Additionally, GBMs are heterogeneous, with a lower mutational burden and reduced PD-L1 expression compared to other responsive tumor types. Although ICI treatment on its own has not significantly extended the survival of patients with GBM, upcoming studies should investigate the potential of combining ICI with various other immunotherapies such as CAR-T, oncolytic viruses, vaccines, or with alternative approaches like stereotactic surgery and localized chemotherapy [8] (Figure 1).

Another promising form of immunotherapy, chimeric antigen receptor (CAR) T cells (CAR-T), have garnered significant attention from researchers and oncologists. In CAR-T therapy, patient blood samples are genetically engineered to express a chimeric antigen receptor (CAR) that can specifically bind to cancer cells. These CAR-T cells are then multiplied and reintroduced into patients to target and destroy cancerous cells. Since 2017, the FDA has approved six CAR-T-cell therapies for treating blood cancers like lymphomas, leukemia, and multiple myeloma [157]. Among these, four target the CD19 antigen (Kymriah, Yescarta, Tecartus, and Breyanzi) [158,159,160,161], while two target the BCMA antigen (Abema and Carvykti) [162,163,164,165].

Currently, most gene therapies directed at GBM are still in the clinical trial phase [21]. The primary tumor-specific antigen in gliomas is a mutated form of EGFR, EGFRvIII, which is found in about 30% of malignant glioma cases. O’Rourke et al., conducted an innovative clinical trial using a type of T cells called autologous T cells that were engineered with a chimeric antigen receptor (CAR) designed to target EGFRvIII [166]. Their phase I/II trial aimed to assess the safety and feasibility of administering these anti-EGFRvIII CAR-T cells to patients with GBM whose tumors express the EGFRvIII molecule. Their results demonstrated that a single infusion of CART-EGFRvIII T cells was both feasible and safe without any adverse effects [166]. In another phase I study (NCT02209376), a single dose of EGFRvIII CAR-T cells was administered to 10 patients. The median OS was around 8 months, with one patient showing stable disease for over 18 months [166]. All patients exhibited detectable proliferation of CAR-T-EGFRvIII cells in the peripheral blood for up to 30 days post-infusion, after which they became undetectable via flow cytometry [166]. In a small open-label study, three individuals with recurrent GBM underwent therapy with CARv3-TEAM-E T cells. These cells were engineered to target both EGFR variant III and wild-type EGFR proteins. The treatment demonstrated a lack of severe adverse events beyond grade 3 or any dose-limiting toxic effects. Although there was notable and swift radiographic regression of the tumor following a single intraventricular infusion, this response turned out to be temporary for two of the three participants [167]. Limitations of EGFRvIII CAR-T therapy include antigen escape and the heterogeneous expression of EGFRvIII in GBM [168].

Furthermore, the human epidermal growth factor receptor 2 (HER2), present in up to 80% of GBM tumors, has been a target for CAR-T-cell therapy in patients with malignant glioma [169,170]. A phase I study established the optimal dose of HER2 chimeric antigen receptor-expressing cytomegalovirus (CMV)-specific cytotoxic T cells in patients with progressive GBM (HERT-GB). Scientists modified the HER2 antibody to bind to T cells and discovered that adding the CD28 protein to the HER2 chimeric receptor (HER2-CAR) boosted T-cell activity [169]. The phase I clinical trial showed that intravenous infusion of virus-specific CAR-T cells in patients with GBM is safe, with potential clinical advantages, prompting further investigation in a phase IIb study [169].

Like other cancer treatments, immunotherapy can lead to significant side effects due to the immune system’s heightened activity against both tumor cells and healthy tissues [151]. Patients may experience flu-like symptoms, such as fever, chills, and fatigue, along with swelling, headaches, and heart palpitations. Combining adjuvant therapies with standard treatments like chemotherapy for GBM and other brain tumors shows promise in overcoming the challenges of the tumor microenvironment and immunosuppression, potentially enhancing treatment efficacy by targeting multiple pathways, reducing drug dosages, and improving immune system responses. However, further research is needed to fully understand the synergistic effects and optimize these treatment strategies for better outcomes (Figure 1).

### 7.2. Oncolytic Virotherapy

Managing patients with gliomas becomes more complex due to the requirement of systemically administered substances to penetrate the BBB and the necessity to prevent excessive inflammation, which could result in significant neurological impairments. Despite these challenges, gliomas typically remain localized in the CNS. Thus, localized strategies like oncolytic viruses (OVs) present an interesting treatment avenue because of the localized tumor growth and the susceptibility of glioma cells to viral infections, such as those caused by HSV1, adenoviruses, and polioviruses [171]. Oncolytic virotherapy is an innovative approach that shows promising potential for the treatment of brain tumors [143,171,172]. This system involves using genetically modified viruses to target and destroy cancer cells while leaving healthy cells unharmed. One of the key advantages of oncolytic virotherapy is its ability to specifically target tumor cells, which can be challenging with traditional treatments like chemotherapy and radiation therapy that often affect healthy cells as well [171].

In brain tumor treatment, oncolytic virotherapy offers several unique benefits. Firstly, the BBB, which typically hinders the delivery of therapeutic agents to the brain, can be circumvented by using viral vectors that can cross this barrier [153]. This means that the OVs can directly reach the tumor site, enhancing their effectiveness in killing cancer cells within the brain. Another advantage of oncolytic virotherapy in brain tumor treatment is its potential in personalized medicine [173]. Scientists can tailor the viral vectors to target specific molecular markers or mutations present in individual patients’ tumors. This customization increases the therapy’s precision and reduces the risk of off-target effects, leading to better treatment outcomes and fewer side effects for patients. Furthermore, oncolytic virotherapy can be combined with other treatment modalities, such as immunotherapy, to create synergistic effects [174]. By harnessing the immune system’s ability to recognize and attack cancer cells, this combination approach can enhance the overall antitumor response and improve long-term survival rates in patients with brain tumors [8,174]. DNX-2401, an oncolytic adenovirus, is designed to replicate selectively in cancer cells with certain genetic defects. It has features like a 24-base pair deletion in the E1A gene for this selectivity and an RGD peptide insertion to enhance infectivity in glioblastoma cells [175]. When delivered into tumors, it triggers an immune response marked by increased T-cell presence and changes in checkpoint protein expression [176,177]. Combining DNX-2401 with immune checkpoint blockade, specifically pembrolizumab, led to promising outcomes in patients with recurrent glioblastoma, with a median survival of 12.5 months and over half experiencing clinical benefits [175]. Notably, a few patients showed durable responses beyond 45 months.

Teserpaturev (G47Δ) is a third-generation HSV1-based oncolytic virus (OV) approved in Japan for treating malignant gliomas [172]. A single-arm phase II trial showed an 84.2% one-year OS rate in patients with recurrent and/or residual GBM, with manageable adverse events like fever, vomiting, nausea, and leukopenia being reported [178]. Another promising agent, the oncolytic HSV1 strain G207, demonstrated efficacy in pediatric patients with high-grade gliomas, resulting in mostly grade 1 adverse events and notable responses in nearly all patients, with a median OS of 12.2 months [179,180]. In a phase I trial, 41 patients with recurrent GBM were injected with CAN-3110, an oncolytic herpes virus (oHSV) [181,182]. Unlike other oHSVs, CAN-3110 contains the viral neurovirulence ICP34.5 gene controlled by a nestin promoter. Nestin, overexpressed in GBM and invasive tumors but not in healthy brain tissue, enables CAN-3110 to replicate preferentially in tumors. The trial encountered manageable toxicities. Patients with positive HSV1 serology showed improved survival and better clearance of CAN-3110 from tumors. Survival was linked to changes in T-cell counts and diversity, specific T-cell clonotype expansion/contraction, and immune activation signatures in tumors [181]. These findings validate that intralesional oHSV treatment boosts anticancer immune responses, especially in patients with prior exposure to the injected virus (NCT03152318).

In addition to DNA viruses like HSV1, RNA viruses such as poliovirus-based Ovs are undergoing clinical trials for glioma treatment. To address the safety concerns associated with unmodified poliovirus strains, a recombinant attenuated poliovirus (PVS-RIPO) has been developed, showing promising results in a phase II study of recurrent glioblastoma patients, albeit with some grade 3–5 adverse events [183,184]. PVSRIPO targets tumor cells by engaging the poliovirus receptor CD155. A phase I trial showed that intratumoral PVSRIPO treatment in patients with recurrent GBM led to improved OS compared to historical controls [183]. Current clinical investigations include a phase II trial (NCT02986178) evaluating PVSRIPO as a standalone therapy, along with phase I/II (NCT03973879) and phase II (NCT04479241) trials assessing the combination of PVSRIPO with either anti-PD-L1 atezolizumab or anti-PD1 pembrolizumab, respectively. In another phase II trial involving patients with recurrent GBM (NCT00870181), participants received an intraarterial cerebral infusion of AdV-TK followed by ganciclovir and mannitol to disrupt the BBB. The results showed a significantly longer median PFS in the AdV-TK group (34.9 weeks) compared to the control group (7.4 weeks) (*p* < 0.001) [185]. Additionally, the median OS was 45.7 weeks in the AdV-TK group, in contrast to 8.6 weeks in the controls (*p* < 0.001).

Despite these strides, challenges persist in optimizing the efficacy and safety of oncolytic virotherapy for brain tumors. Special attention is needed due to OVs being live replicating viruses, posing risks of viral shedding, unintentional transmission, immune clearance, neurotoxicity, and therapy resistance. Nevertheless, ongoing research and clinical trials are refining this approach, offering hope for improved outcomes and new options for patients with brain tumors in the future (Figure 1).

### 7.3. CRISPR/Cas9

CRISPR/Cas9 has emerged as a promising tool for exploring new avenues for the treatment of brain tumors. This revolutionary gene-editing technology offers precision and efficiency in targeting specific genes associated with tumor growth, making it a potentially transformative approach in oncology [186,187]. For example, researchers can use CRISPR/Cas to disrupt genes responsible for promoting cell proliferation or inhibiting apoptosis, thereby slowing down tumor growth and enhancing the effectiveness of traditional therapies like chemotherapy and radiation [186,188].

In one investigational study, CHAF1A was targeted for CRISPR-Cas9 knockout, leading to an examination of its effects on the AKT/FOXO3a/Bim pathway and its influence on proliferation and DNA repair mechanisms crucial to cell cycle regulation [189]. A knockout study involving ATM, PTEN, p85, and XIAP genes revealed their roles as tumor suppressors, contributing to our understanding of the complex regulatory mechanisms that control the cell cycle in GBM [190].

In GBM research, there is a growing focus on deciphering how the tumor microenvironment, particularly concerning angiogenesis, can be intricately adjusted to inform therapeutic approaches. One such method involves targeting the Notch1 gene using a knockdown technique to address issues like hypoxia, angiogenesis, and tumor growth. Notch1 plays a critical role in various cellular processes, and its modulation was aimed at disrupting key pathways associated with angiogenesis, which is a crucial aspect of GBM progression. The study utilized CRISPR/Cas9 technology to decrease Notch1 expression, with the goal of unraveling the complex interplay between hypoxia, angiogenesis, and the overall growth dynamics of GBM malignant cells.

Lu et al. pursued further investigation into angiogenesis-related genes by focusing on the BIG1 and BIG2 genes in 2019 [191]. They targeted VEGF through knockdown techniques. In a separate study, researchers aimed to address CRISPR/Cas9 gene-editing resistance in GBM treatment by specifically targeting the ALDH1A3 gene, which yielded promising results [192]. Targeting ALDH1A3 significantly impacted TMZ resistance, especially at dosages of 300 µM. Furthermore, MGMT knockdown was explored to sensitize GBM cells to TMZ treatment, while MUC1 knockdown shed light on its role in DNA damage repair during chemotherapy and radiation [193].

Moreover, CRISPR/Cas enables personalized medicine by allowing clinicians to tailor treatments based on the genetic profiles of individual patient tumors. By analyzing the unique genetic mutations driving tumor development, healthcare providers can design CRISPR-based therapies that specifically target these aberrations, potentially leading to more effective and less toxic treatment outcomes. However, it is essential to note that while CRISPR/Cas shows great potential, there are still challenges and ethical considerations that need to be addressed. Off-target effects, unintended genetic changes, and immune responses to CRISPR-modified cells are among the factors that researchers are actively working to overcome to ensure the safety and efficacy of CRISPR-based therapies for brain tumor treatment. Additionally, CRISPR/Cas9 technology faces challenges in efficiently delivering its therapeutic payload to brain tumors, often resulting in limited efficacy [194,195,196,197].

## 8. Conclusions

Great strides have been made in understanding GBM at the molecular level, yet integrating these findings into clinical practice faces significant challenges. One major hurdle is the intratumor heterogeneity of GBM, which complicates the treatment outcomes. This heterogeneity, coupled with the complexity of GBM’s molecular landscape, will likely require future combination therapies targeting multiple driver events. Another critical challenge lies in pharmacokinetics, with the blood–brain barrier hindering effective drug distribution within the brain, which is a crucial aspect of GBM treatment. The current management approach for GBM comprises several key steps: maximal safe surgical resection, followed by radiation therapy, and adjuvant treatments like temozolomide and TTFields (Figure 3) [4]. Tailoring the treatment plan involves considering various factors such as tumor characteristics, patient health, and molecular markers. Collaboration among specialists like radiation oncologists, neurosurgeons, and medical physicists is essential for personalized care. The integration of artificial intelligence (AI) for analyzing brain tumor images and refining treatment strategies represents a promising frontier. AI’s role in enhancing tumor genotyping accuracy, delineating tumor volumes, and predicting outcomes underscores its potential in precision medicine, aiding in targeted chemotherapy approaches. Note that while this review aims to summarize effective GBM management, some pertinent reports may not be included.

## Figures and Tables

**Figure 1 biomedicines-12-01376-f001:**
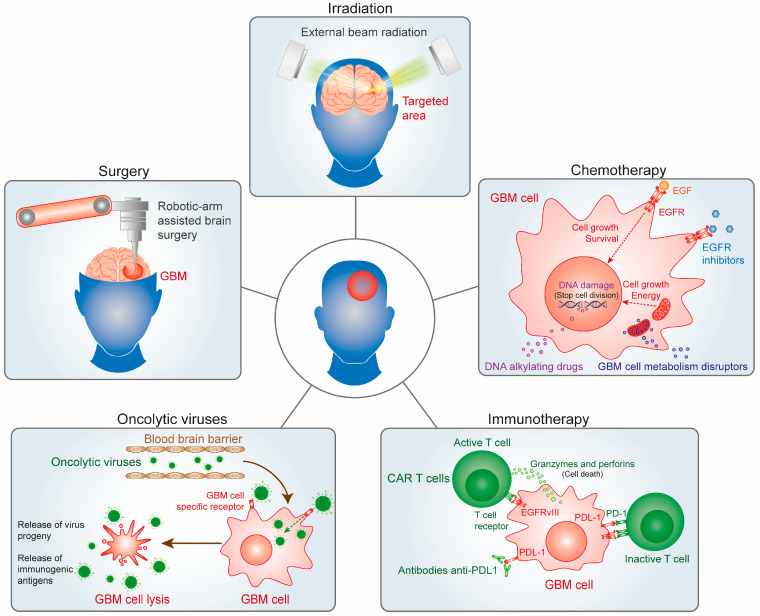
Multimodal approaches to glioblastoma treatment encompass various strategies. Robotic-assisted procedures enhance the effectiveness and accessibility of brain tumor surgery by offering heightened precision and minimal invasiveness. Alongside this, external beam radiation therapy utilizes computer-controlled linear accelerators to administer precise radiation doses to targeted areas within the tumor. Additionally, chemotherapy plays a crucial role by utilizing potent drugs to eliminate rapidly dividing cancer cells, often administered orally or intravenously to complement other treatment modalities like surgery and radiation therapy. Moreover, innovative strategies, such as immunotherapy and oncolytic virus therapy, are being explored to harness the immune system’s capabilities and directly combat cancer cells.

**Figure 2 biomedicines-12-01376-f002:**
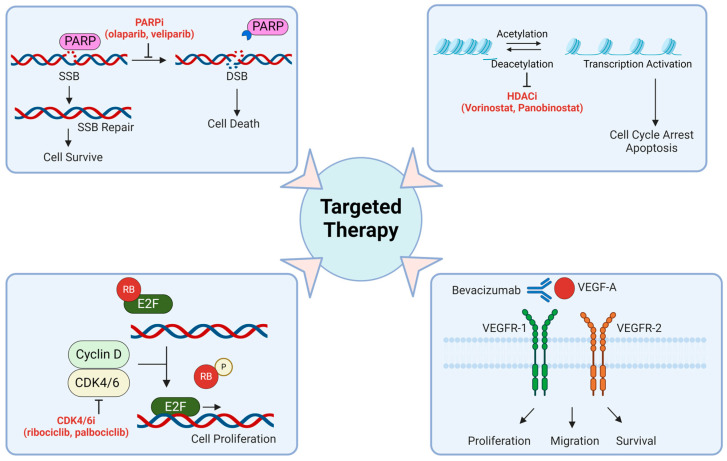
Refining glioblastoma treatment involves employing methods for targeted therapy. **Upper left**: PARP inhibitors target an enzyme called poly ADP-ribose polymerase (PARP), which plays a key role in repairing damaged DNA. SSB: single strand break; DSB: double stand break; PARPi: PARP inhibitor. **Upper right**: HDAC inhibitors target enzymes known as histone deacetylases, which play a role in regulating gene expression. HDACi: HDAC inhibitor. **Lower left**: CDK4/6 inhibitors block the activities of cyclin-dependent kinases 4 and 6, which are involved in cell cycle progression. Cyclin-D-CDK4/6 complexes regulate the cell cycle by phosphorylating RB. Phosphorylated RB releases E2F, facilitating cell cycle progression from the G1 to S phase. CDK4/6i: CDK4/6 inhibitor; RB: Retinoblastoma tumor suppressor gene. **Lower right**: Bevacizumab blocks the action of vascular endothelial growth factor (VEGF) and inhibits angiogenesis, which is vital for tumor growth and metastasis. The figure was generated using BioRender.

**Figure 3 biomedicines-12-01376-f003:**
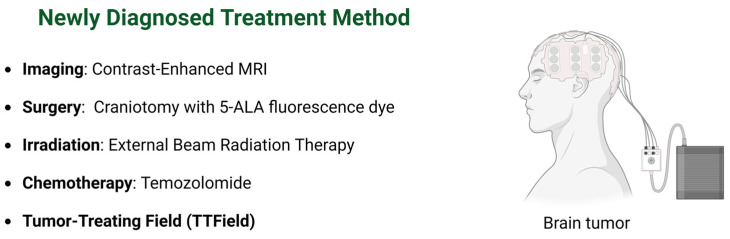
Current treatment for a newly diagnosed brain tumor. Patients will receive maximal safe surgical resection, followed by radiation therapy and adjuvant treatments like temozolomide and TTFields. The figure was generated using BioRender.

## Data Availability

Not applicable.

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
