# Peer review of "Revolutionizing Brain Tumor Care: Emerging Technologies and Strategies"

_biomedicines, 2024, doi:10.3390/biomedicines12061376_

Round 1
Reviewer 1 Report
Comments and Suggestions for Authors
The present study represents a nice compendium about the state of art of GBM management, on the other hand it does not fill any gap on the exisiting body of literature.
Author Response
Thank you for your feedback on our manuscript. We appreciate your acknowledgment of our effort to compile a comprehensive overview of the state of the art in GBM management. The primary objective of our review is to highlight cutting-edge technologies in current and ongoing stages of development. By doing so, we aim to provide readers with valuable insights to navigate their choices in brain tumor treatment. While our review may not introduce entirely new findings to the existing body of literature, it serves as a crucial resource for both clinicians and patients by consolidating the latest advancements and emerging trends in GBM management. This approach helps bridge the knowledge gap between ongoing research developments and practical, patient-centered applications. We believe that this synthesis of information will support informed decision-making and enhance patient outcomes by making the latest technological advancements more accessible to the broader medical community and those affected by GBM.
Reviewer 2 Report
Comments and Suggestions for Authors
The review paper “Revolutionizing Brain Tumor Care: Emerging Technologies 2 and Strategies” by Trang T.T and collaborators deals with emerging combined approaches in the therapy of glioblastoma, one of the most aggressive tumors with a poor prognosis. In fact, standard care for newly diagnosed GBM patients has remained unchanged for many years and typically involves surgical resection, followed by a combination of chemotherapy using temozolomide (TMZ) and radiotherapy. However, recurrence often occurs within several months despite these treatments. In this review work the authors describe current and emerging treatment methods for this type of tumor, highlighting the advantages and limitations of the possible therapeutic options currently available to patients.
General comments
The paper, as a whole, is interesting and offers a broad overview especially of the instrumental options available to surgeons in dealing with this pathology. However, I think that adding some information to what has been reported can help the reader to better frame the topic, especially to understand which options are already widely used in common practice and therefore actually available to the common patient who is not included in particular clinical trials or experimental studies.
Specific poins
- Introduction. For the reader who does not specifically study glioblastoma, it would be useful to include a table in which the molecular characteristics are associated with the prognosis and elective treatments.
- Reading the work, it is not clear how both radiological and surgical techniques described are currently implemented. The authors should add quantitative data referring to at least some centers specialized in glioblastoma therapy to which the authors refer. In other words: in what percentage of interventions are the techniques described used? As far as I know, some are currently only experimental and not used in daily clinical practice.
- Reading the work, I seem to understand that even the most selective and cutting-edge techniques do not produce significant effects on either survival or the patient's quality of life? Did I get it right? If not, emphasize the improvements. Furthermore, was a greater benefit of one or the other irradiation technique observed in men compared to women and/or based on a different tumor location?
- There are many works that refer to neuroleptic drugs capable of crossing the blood-brain barrier as potentially interesting and targeted for use in association with TMZ. Could the authors add some considerations on these therapeutic opportunities and on any ongoing trials on these drugs considered repositionable?
Author Response
We thank the reviewer for the appreciation and careful examination of our work.
Specific points
Introduction. For the reader who does not specifically study glioblastoma, it would be useful to include a table in which the molecular characteristics are associated with the prognosis and elective treatments.
Response: As of now, the methylation status of the MGMT promoter is the only reliable predictor of response to standard of care drugs like Temozolomide. While EGFR and VEGF inhibitors hold promise for recurrent GBM, their effectiveness is still under thorough clinical evaluation. In response to this point, we have now added the following material to the review: “Patients with MGMT methylation exhibit improved median OS rates compared to those without it: 21.7 months with chemotherapy and radiation versus 15.3 months with radiation alone for methylated patients; 12.7 months with chemoradiation versus 11.8 months with radiation alone for unmethylated patients. Thus, MGMT promoter methylation is a crucial predictor for TMZ chemotherapy efficacy, especially in personalizing treatments for elderly GBM patients” (Manuscript page 9).
Reading the work, it is not clear how both radiological and surgical techniques described are currently implemented. The authors should add quantitative data referring to at least some centers specialized in glioblastoma therapy to which the authors refer. In other words: in what percentage of interventions are the techniques described used? As far as I know, some are currently only experimental and not used in daily clinical practice.
Response: This is an excellent question, although currently we do not have data regarding the specific percentage of interventions utilizing described techniques in both radiological and surgical settings. Patients are encouraged to consult with neuro-oncologists in their vicinity to inquire about the available facilities in each hospital to support their treatments.
Reading the work, I seem to understand that even the most selective and cutting-edge techniques do not produce significant effects on either survival or the patient's quality of life? Did I get it right? If not, emphasize the improvements. Furthermore, was a greater benefit of one or the other irradiation technique observed in men compared to women and/or based on a different tumor location?
Response: While it is true that some advanced techniques might have limited impact on long-term survival for certain conditions, they often bring significant improvements in other areas, such as reducing tumor size, prolonging periods of remission, and enhancing the patient's day-to-day life.
We have added the following material to respond to this point: “Research has established a disparity in glioblastoma (GBM) survival based on sex, where females exhibit a notable survival advantage. The incidence of GBM is higher in women's right temporal lobe, while in men, the left temporal lobe is more affected. A large study conducted by Dr. Barnholtz-Sloan's group at the National Cancer Institute found no sex-based survival differences among individuals with MGMT promoter unmethylated and those with MGMT promoter methylated who received either TMZ and radiation or TMZ alone. However, a significant survival gap was observed among individuals with MGMT promoter methylated, particularly with females showing significantly lower survival rates than males when treated with radiation alone (females: 13.4 months vs. males: 24.5 months, p=0.01).” (Manuscript page 9, lines 394-403).
There are many works that refer to neuroleptic drugs capable of crossing the blood-brain barrier as potentially interesting and targeted for use in association with TMZ. Could the authors add some considerations on these therapeutic opportunities and on any ongoing trials on these drugs considered repositionable?
Response: Neuroleptic drugs are used adjunctively to manage specific symptoms related to the tumor's effects on mental health rather than to treat the tumor itself. These medications work by affecting neurotransmitters like dopamine and serotonin in the brain, which can help alleviate symptoms of psychosis. We decided not to include such drugs in this review article.
Reviewer 3 Report
Comments and Suggestions for Authors
Regarding the challenges in diagnosis and therapy of glioblastoma multiforme (GBM), especially the difficulty in drug delivery through the blood brain barrier, this review highlights advanced technologies with great potential in GBM imaging and treatment, thus hoping to provide patients with valuable insights into brain tumor treatments. The manuscript is well-organized and therefore I would recommend the publication of this paper after minor revision:
1. Magnetic Resonance Imaging is mainly discussed in “Imaging” section. What about other imaging methods such as CT or PET? Please provide some comparison between these imaging technologies.
2. It is recommended to provide a table or figure displaying the advantages and disadvantages of various irradiation therapeutic strategies.
3. Some novel therapies are introduced in the manuscript such as TTFields therapy. Is this therapeutic strategy approved by FDA? If not, what is the main challenge?
4. Please read the following works on tumor therapy that are closely related to this research: “Kuikun Yang,# Guocan Yu,# Zhiqing Yang, Ludan Yue, Xiangjun Zhang, Chen Sun, Jianwen Wei, Lang Rao, Xiaoyuan Chen* and Ruibing Wang*, Supramolecular Polymerization-induced Nanoassemblies for Self-augmented Cascade Chemotherapy and Chemodynamic Therapy of Tumour. Angew. Chem. Int. Ed. 2021, 60, 17570.; Qingfu Wang, Chen Zhang, Ya Zhao, Yifan Jin, Shen Zhou, Junde Qin, Wenxin Zhang, Ying Hu,* Xiaoyuan Chen* and Kuikun Yang*, Polyprodrug nanomedicine for chemiexcitation-triggered self-augmented cancer chemotherapy and gas therapy. Biomaterials. 2024, 309, 122606.”
Author Response
We thank the reviewer for the appreciation and careful examination of our work.
- Magnetic Resonance Imaging is mainly discussed in “Imaging” section. What about other imaging methods such as CT or PET? Please provide some comparison between these imaging technologies.
Response: CT scans are beneficial for detecting calcifications, bone involvement, and acute hemorrhage, though they are less commonly used as the primary imaging modality for glioblastoma. PET scans, while not typically employed for the initial diagnosis of glioblastoma, are valuable for assessing tumor metabolism and distinguishing between tumor recurrence and radiation necrosis. Consequently, this review article focuses on the more prevalent use of MRI in glioblastoma diagnosis and monitoring, and does not include detailed discussions on CT and PET scans.
- It is recommended to provide a table or figure displaying the advantages and disadvantages of various irradiation therapeutic strategies.
Response: We have included a table that outlines the benefits and drawbacks of different irradiation therapeutic approaches in response to your request in the manuscript (Manuscript page 8).
- Some novel therapies are introduced in the manuscript such as TTFields therapy. Is this therapeutic strategy approved by FDA? If not, what is the main challenge?
Response: TTFields is a therapeutic strategy approved by FDA for recurrent GBM in 2011 and newly diagnosed GBM in 2015 (Manuscript page 13).
- Please read the following works on tumor therapy that are closely related to this research: “Kuikun Yang,# Guocan Yu,# Zhiqing Yang, Ludan Yue, Xiangjun Zhang, Chen Sun, Jianwen Wei, Lang Rao, Xiaoyuan Chen* and Ruibing Wang*, Supramolecular Polymerization-induced Nanoassemblies for Self-augmented Cascade Chemotherapy and Chemodynamic Therapy of Tumour. Angew. Chem. Int. Ed. 2021, 60, 17570.; Qingfu Wang, Chen Zhang, Ya Zhao, Yifan Jin, Shen Zhou, Junde Qin, Wenxin Zhang, Ying Hu,* Xiaoyuan Chen* and Kuikun Yang*, Polyprodrug nanomedicine for chemiexcitation-triggered self-augmented cancer chemotherapy and gas therapy. Biomaterials. 2024, 309, 122606.”
Response: Thank you for introducing the new method of suppressing tumor growth using CO prodrug-loaded nanomedicine. However, this topic, while very interesting, does not align with the focus of our review paper.